# Phytoplankton Response to Increased Nickel in the Context of Ocean Alkalinity Enhancement

Xiaoke Xin, Giulia Faucher, Ulf Riebesell

GEOMAR Helmholtz Centre for Ocean Research Kiel, Kiel, Germany

*Correspondence to*: Xiaoke Xin (xxin@geomar.de)

**Abstract**

Ocean alkalinity enhancement (OAE) is considered one of the most promising approaches to actively remove carbon dioxide ($CO_2$) from the atmosphere by accelerating the natural process of rock weathering. This approach involves introducing alkaline substances sourced from natural mineral deposits such as olivine, basalt, and carbonates or obtained from industrial waste products such as steel slags, into seawater and dispersing them over coastal areas. Some of these natural and industrial substances contain trace metals, which would be released into the oceans along with the alkalinity enhancement. The trace metals could serve as micronutrients for marine organisms at low concentrations, but could potentially become toxic at high concentrations, adversely affecting marine biota. To comprehensively assess the feasibility of OAE, it is crucial to understand how the phytoplankton, which forms the base of marine food webs, responds to ocean alkalinization and associated trace metal perturbations. As one of the most abundant metals in OAE source materials, understanding the impacts of nickel (Ni) on the phytoplankton is critical for OAE assessment. In this study, we investigated the influence of nickel (Ni) on three representative phytoplankton species over a gradient of nine Ni concentrations (from 0 to 100 µmol L$^{-1}$ with 12 µmol L$^{-1}$ synthetic organic ligand). The impacts of elevated Ni varied among the tested phytoplankton species. The coccolithophore *Emiliania huxleyi* and the dinoflagellate *Amphidinium carterae* exhibited a growth rate inhibition of about 30% and 20%, respectively, at the highest Ni concentrations. The half maximal inhibitory concentration (IC50, at which the growth rate is inhibited by 50%) of both species exceeded the tested range of Ni. This suggests that both species were only mildly affected by the elevated Ni concentrations. In contrast, the diatom *Thalassiosira weissflogii* displayed a considerably higher sensitivity to Ni, with a 60% growth rate inhibition at the highest Ni concentration and an IC50 value of 63.9 µmol L$^{-1}$. In conclusion, the variability in phytoplankton sensitivity to Ni exposure suggests that for OAE applications with Ni-rich materials caution is required and critical toxic thresholds for Ni must be avoided.

## 1 Introduction

The progressive release of anthropogenic carbon dioxide ($CO_2$) into the atmosphere since the industrial revolution resulted in a multitude of environmental challenges, including global warming, ocean acidification, ecosystem alteration, increasing frequency of extreme climatic events, and food insecurity (Pörtner et al., 2022). To keep the effects of climate change within acceptable limits, about 1 to 15 $GtCO_2$ $yr^{-1}$ must be captured by 2100 (Rogelj et al., 2018). Carbon capture activities, known as negative emission technologies (NETs), have moved to the limelight of discussion as they will need to be implemented over the next two decades to meet the climate targets and limit global warming to $< 2°C$ (Shepherd, 2009; Allan et al., 2021). One of the promising NETs is to accelerate natural rock weathering by introducing finely ground alkaline products on land (Enhanced Weathering, EW) or into the surface ocean (Ocean Alkalinity Enhancement, OAE) to remove $CO_2$ from the atmosphere (Minx et al., 2018; Bach et al., 2019). OAE, in addition to enhancing the buffering capacity of seawater, has the co-benefit of mitigating ocean acidification at the deployment sites (Köhler et al., 2010). Natural alkaline minerals, as well as by-products from industrial activity, are potential candidates for EW and OAE (Taylor et al., 2016; Renforth, 2019). Among the most recognized alkaline minerals, olivine rocks have gained considerable attention due to their relatively fast weathering rate, wide availability, and low cost (Schuiling and Krijgsman, 2006; Hartmann et al., 2013). These rocks contain high amounts of trace metals, e.g., nickel (Ni) and chromium (Cr) (Montserrat et al., 2017; Amann et al., 2020) that through OAE could affect coastal and off-shore systems, possibly influencing marine communities (Gaillardet et al., 2003).

In seawater, Ni is present at low concentrations (Donat et al., 1994; Mackey et al., 2002; Saito et al., 2004) and acts as a micronutrient when urea serves as the nitrogen source (Muyssen et al., 2004; Egleston and Morel, 2008). However, this metal at elevated concentrations may emerge as a concern to marine ecosystems, due to its toxicity, bioaccumulation, and biogeochemical cycling (Sclater et al., 1976; Hall and Anderson, 1995; Horvatić and Peršić, 2007; Debelius et al., 2011; DeForest and Schlekat, 2013; Martínez-Ruiz and Martínez-Jerónimo, 2015; Karthikeyan et al., 2018). In olivine-based OAE scenarios, Ni concentrations could rapidly rise above critical levels that become harmful to marine organisms (Montserrat et al., 2017; Hartmann et al., 2023). To date, studies on high Ni concentrations are scarce. In this study, we examined the impacts of Ni on three representative marine phytoplankton species: the diatom *Thalassiosira weissflogii*, the dinoflagellate *Amphidinium carterae*, and the coccolithophore *Emiliania huxleyi*. These species were selected as they represent three

dominant functional groups of phytoplankton. This research aimed to (1) investigate how
different phytoplankton species respond to a gradient of Ni concentrations and (2) compare the
inter-species differences in Ni sensitivity.

## 2 Materials and methods

### 2.1 Strain and culture conditions

Experiments were conducted with cultures of the marine diatom *Emiliania huxleyi* B92/11
(Plymouth Marine Laboratory), the dinoflagellate *Amphidinium carterae* CCAP1102
(University of Oldenburg), and the coccolithophore *Thalassiosira weissflogii* CCMP1336
(Bigelow Laboratory for Ocean Sciences). Algae were cultivated in sterile-filtered (0.2 µm) f/2
media prepared with artificial seawater (Guillard and Ryther, 1962; Kester et al., 1967). Media
were enriched with essential trace metals buffered by ethylenediaminetetraacetic acid (EDTA,
12 µmol $L^{-1}$). Cells grew at 18°C with a 12:12h light and dark cycle under 200 µmol photons
$m^{-2}$ $s^{-1}$ of photosynthetically active radiation (PAR). Media were acclimated to the incubation
temperature prior to inoculation from the precultures to avoid a potential thermal shock.

### 2.2 Experimental setup

To determine the toxicity of nickel, a stock solution (as $NiCl_2 \times 6H_2O$) was prepared, with a
nominal value of 50 mmol $L^{-1}$. All bottles were soaked with 10 % HCl (Fisher) for 24 h and
rinsed with Milli-Q water before the experiment. Stock solutions of Ni were added to the algal
media for different Ni concentration treatments (0.01, 0.1, 1, 5, 10, 20, 50, and 100 µmol $L^{-1}$)
and the control with exclusive f/2 medium. Experiments were performed in triplicate
independent 75 mL falcon flasks for each concentration. All cultures were gently turned by
hand twice a day to avoid cells from settling. Samples were always collected at the same time
of day between 9:00 a.m. and 10:00 a.m. to avoid an effect of the photocycle. Samples (1 mL)
of each flask were collected into sterile 2 mL microtubes. The cell density was determined
daily with a flow cytometer (BD Accuri™ C6). To minimize the impact of changes in the
carbonate chemistry of the medium induced by cellular metabolism, the phytoplankton biomass
at the harvest time should consume less than 5% of the total dissolved inorganic carbon
(Zondervan et al., 2002). The maximum density was determined based on the cell carbon quota
of the tested species. Accordingly, the maximum cell densities of *A. carterae, T. weissflogii*,
and *E. huxleyi* never exceeded 12000 cells $mL^{-1}$, 15000 cells $mL^{-1}$, and 130000 cells $mL^{-1}$,
respectively (Zondervan et al., 2002; Olenina et al., 2006).
**2.3 Growth rate and IC50 value determination**
Growth rates ($\mu$; d$^{-1}$) were calculated from cell density following:
$$\mu = \frac{ln(c_f) - ln(c_0)}{d},$$ (1)
where $c_0$ and $c_f$ are the cell densities at the beginning and end of the experiment,
respectively; $d$ is the duration of the experiment.
The toxic response was expressed as:
$$I = \left(1 - \frac{\mu_{inhibited}}{\mu_{control}}\right) \times 100\%,$$ (2)
where $I$ is the growth inhibition and $\mu$ is the growth rate.
Dose-response curves were constructed for growth rate following Stephenson et al. (2000).
Nonlinear regression models were determined by the least square method. Model equations
were chosen based on scatter plots of the growth rates of the different species. The sigmoidal
model was applied as:
$$Y = \frac{t}{1 + (\frac{C}{u})^B},$$ (3)
where $Y$ is the growth rate and $C$ is the Ni concentration. The parameter $t$ is the control response,
and $u$ and $B$ define the location and shape of the equation, respectively. The half maximal
inhibitory concentration (IC50), at which the growth rate is inhibited by 50%, was determined
from the dose-response curve.
**2.4 Nickel measurement**
At the end of the growth experiment, 30 mL media of each sample were filtered with 0.2 µm
sterile disc filters to remove the algae. The filtered media were collected for Ni measurements.
The total nickel concentrations were measured with ThermoFisher Scientific ElementXR to
ensure that the target concentrations were reached. The measured Ni concentrations were used
as input data for the program Visual Minteq 3.1 (Gustafsson, 2013). The software utilizes a
chemical equilibrium model to calculate metal speciation and obtain the concentrations of free
$Ni^{2+}$.
**2.5 Statistical analysis**
Data represent means ± standard deviations (N = 3). ANOVA was performed on the cell density
and growth rates to assess the effect of Ni concentrations. Differences among treatments were
tested with Tukey's HSD ANOVA test. Significant differences were reported at the 95 %

confidence level. All statistics were conducted in the Rstudio environment (R packages

"tidyverse" and "ggplot2"; Posit team, 2023; R Core Team, 2023).

**3 Result**

**3.1 Ni concentrations**

Table 1. Target, measured, and free concentrations of Ni for the different phytoplankton cultures at the end of the experiment. SD standard deviation.

| | Target Ni concentration ($\mu$mol/L) | Measured Ni concentration ($\mu$mol/L) | Measured Ni SD ($\mu$mol/L) | Free $Ni^{2+}$ concentration ($\mu$mol/L) | Free $Ni^{2+}$ SD ($\mu$mol/L) |
|---|---|---|---|---|---|
| Stock | $5 \times 10^4$ | $4.16 \times 10^4$ | | | |
| | 0 | 0.00 | 0 | 0 | 0 |
| | 0.01 | 0.01 | $3.45 \times 10^{-3}$ | $8.0 \times 10^{-6}$ | $7.83 \times 10^{-6}$ |
| | 0.1 | 0.05 | 0.02 | $7.94 \times 10^{-5}$ | $2.87 \times 10^{-5}$ |
| | 1 | 0.70 | 0.02 | $1.61 \times 10^{-3}$ | $7.60 \times 10^{-5}$ |
| *A. carterae* | 5 | 3.59 | 0.41 | 0.03 | $6.95 \times 10^{-3}$ |
| | 10 | 7.04 | 0.54 | 0.17 | 0.05 |
| | 20 | 16.19 | 0.39 | 4.47 | 0.28 |
| | 50 | 40.59 | 0.97 | 22.26 | 0.71 |
| | 100 | 80.57 | 2.03 | 51.64 | 1.49 |
| | 0 | 0.00 | 0 | 0 | 0 |
| | 0.01 | / | / | / | / |
| | 0.1 | 0.12 | 0.02 | $9.17 \times 10^{-5}$ | $2.56 \times 10^{-5}$ |
| | 1 | 0.88 | 0.07 | $2.21 \times 10^{-3}$ | $2.53 \times 10^{-4}$ |
| *E. huxleyi* | 5 | 3.89 | 0.31 | 0.03 | $5.71 \times 10^{-3}$ |
| | 10 | 7.32 | 0.21 | 0.19 | 0.02 |
| | 20 | 15.34 | 2.09 | 3.90 | 1.43 |
| | 50 | 40.17 | 0.83 | 21.95 | 0.61 |
| | 100 | 78.60 | 2.92 | 50.19 | 2.14 |
| | 0 | 0.00 | 0 | 0 | 0 |
| | 0.01 | 0.07 | / | $3.03 \times 10^{-6}$ | / |
| | 0.1 | 0.07 | $1.36 \times 10^{-3}$ | $9.31 \times 10^{-5}$ | $2.19 \times 10^{-6}$ |
| | 1 | 0.77 | $3.99 \times 10^{-3}$ | $1.81 \times 10^{-3}$ | $1.33 \times 10^{-5}$ |
| *T. weissflogii* | 5 | 3.99 | 0.13 | 0.03 | $2.42 \times 10^{-3}$ |
| | 10 | 8.01 | 0.13 | 0.27 | 0.02 |
| | 20 | 16.50 | 0.27 | 4.68 | 0.19 |
| | 50 | 40.91 | 0.68 | 22.48 | 0.50 |
| | 100 | 77.80 | 0.76 | 49.60 | 0.56 |

The stock solution was not acidified to avoid pH changes in the culture media. For this reason,

nickel carbonate precipitation occurred in the stock solution and approximately 80 % of the

target Ni concentrations were achieved (Table 1, Fig. S1; Gad 2023). Free $Ni^{2+}$ was chelated

by ligand at low concentrations but concentrations of free $Ni^{2+}$ increased with elevated total Ni

(Table 1, Fig. 1). Ligand chelated more $Ni^{2+}$ with elevated total Ni concentration, while the

binding ability decreased. More than 60 % free $Ni^{2+}$ were beyond ligand binding capacity at the highest Ni concentration.

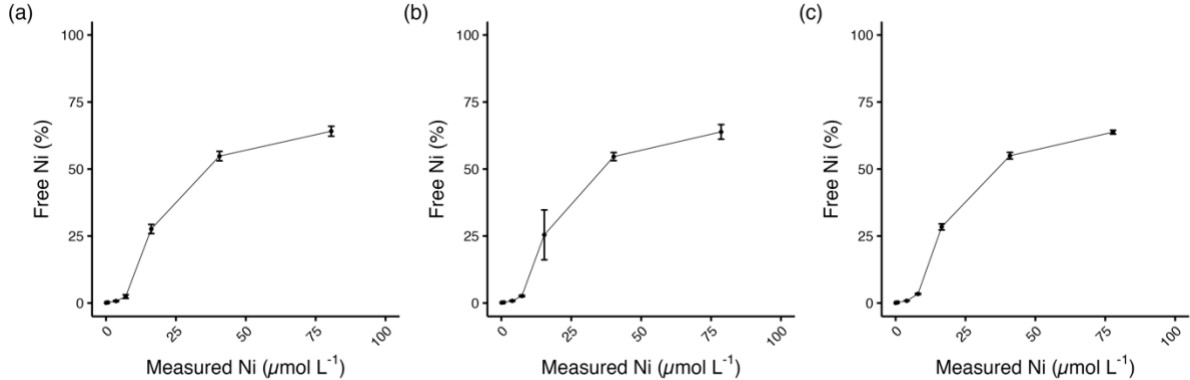

Fig. 1. Percentage of free $Ni^{2+}$ in (**a**) *A. carterae*, (**b**) *E. huxleyi*, and (**c**) *T. weissflogii* cultures at the end of the experiment. Error bars denote standard deviations (N = 3).

**3.2 Growth response and cell density accumulation**

All three species survived the highest tested concentrations. With increasing Ni, the cell densities and growth rates of the three species decreased, albeit differently.

**3.2.1** *Amphidinium carterae*

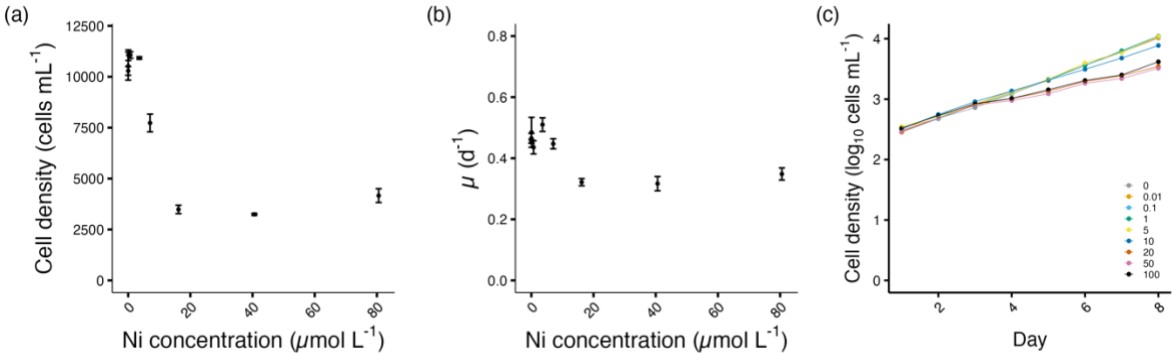

Fig. 2. Growth performance of *A. carterae*. (**a**) Cell densities (cells $mL^{-1}$), and (**b**) Growth rates ($d^{-1}$) plotted against measured Ni concentrations on the final experimental day. (**c**) Log-transformed cell densities plotted against time (day) to indicate the time of response; target Ni concentrations ($\mu$mol $L^{-1}$) used for clarity. Error bars denote standard deviations (N = 3). If not visible, error bars are smaller than symbols.

The cell densities of *A. carterae* decreased significantly from 7.1 $\mu$mol $L^{-1}$ and the growth rates from 16.2 $\mu$mol $L^{-1}$ Ni concentrations ($p < 0.05$; Fig. 2a, Fig. 2b). Growth was not inhibited until day 4 after the exposure to Ni (Fig. 2c). From 16.2 to 80.6 $\mu$mol $L^{-1}$ Ni concentrations, we observed the maximum decrease in cell density of about 59–66 %, with the growth rate decreasing up to 30 % at the highest Ni concentration compared to the control (Fig. 2a, Fig. 2b, Table S1).

### 3.2.2 *Emiliania huxleyi*

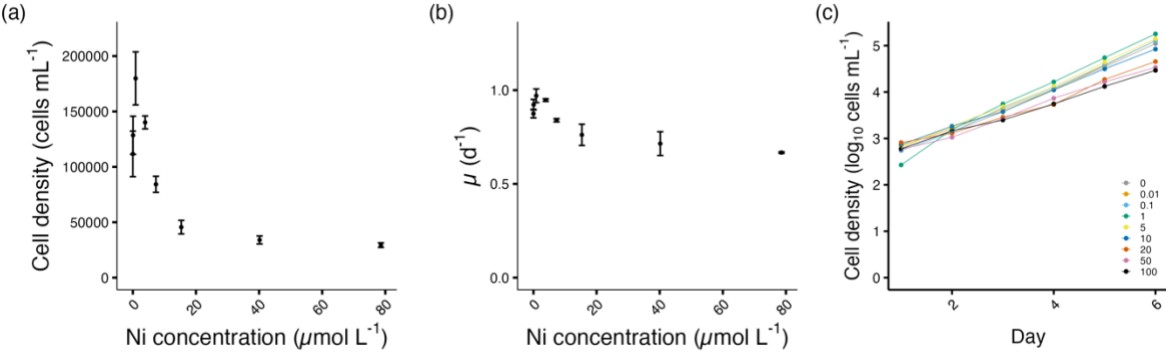

Fig. 3. Growth performance of *E. huxleyi*. (**a**) Cell densities (cells mL$^{-1}$), and (**b**) Growth rates (d$^{-1}$) plotted against measured Ni concentrations on the final experimental day. (**c**) Log-transformed cell densities plotted against time (day) to indicate the time of response; target Ni concentrations (µmol L$^{-1}$) were used for clarity. Error bars denote standard deviations (N = 3). Note that the data point at 0.01 µmol L$^{-1}$ was excluded in (a) and (b) due to technical measurement error. If not visible, error bars are smaller than symbols.

The cell densities and growth rates of *E. huxleyi* increased with the addition of Ni up to 3.89 µmol L$^{-1}$. At 0.9 µmol L$^{-1}$ Ni, the maximum cell density with a 63 % increase was observed ($p$ < 0.01). The growth rate increased by about 11 % but this value is not statistically significant ($p$ = 0.07).

From 15.3 to 78.6 µmol L$^{-1}$, the cell densities decreased significantly between 57–72 % compared to the control ($p$ < 0.05; Fig. 3a, Table S1). The decrease in growth rate reached up to 24 % at the highest Ni concentration compared to the control (Fig. 3b, Table S1). The growth variance of *E. huxleyi* started on day 3 after being exposed to Ni (Fig. 3c).

### 3.2.3 *Thalassiosira weissflogii*

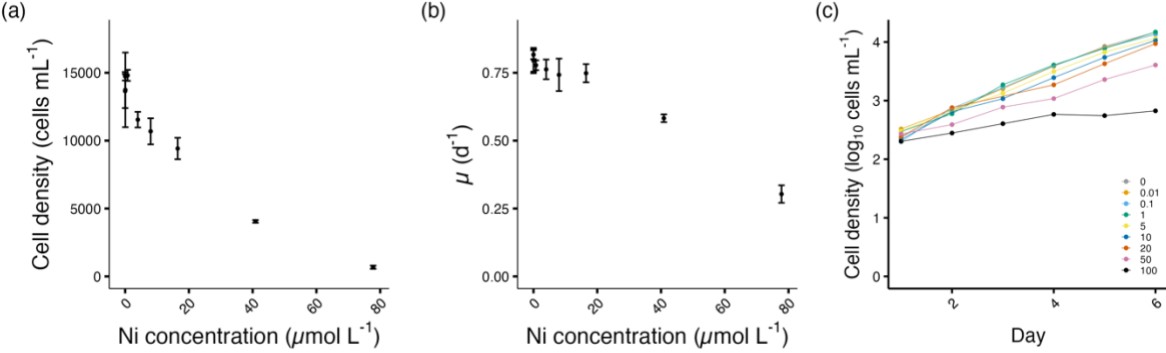

Fig. 4. Growth performance of *T. weissflogii*. (**a**) Cell densities (cells mL$^{-1}$), and (**b**) Growth rates (d$^{-1}$) plotted against measured Ni concentrations on the final experimental day. (**c**) Log-transformed cell densities plotted against time (day) to indicate the time of response; target Ni concentrations (µmol L$^{-1}$) used for clarity. Error bars denote standard deviations (N = 3). If not visible, error bars are smaller than symbols.

The cell densities of *T. weissflogii* remained relatively stable until 8.0 µmol L$^{-1}$ Ni, above which
the densities started to decrease significantly ($p < 0.05$; Fig. 4a, Table S1). Similarly, the growth
rates of *T. weissflogii* remained relatively stable until 40.9 µmol L$^{-1}$ Ni, above which the growth
rates started to decrease significantly ($p < 0.05$; Fig. 4b, Table S1). After being exposed to Ni,
*T. weissflogii* reacted immediately from day 2 onwards (Fig. 4c).

**3.3 Determination of IC50**

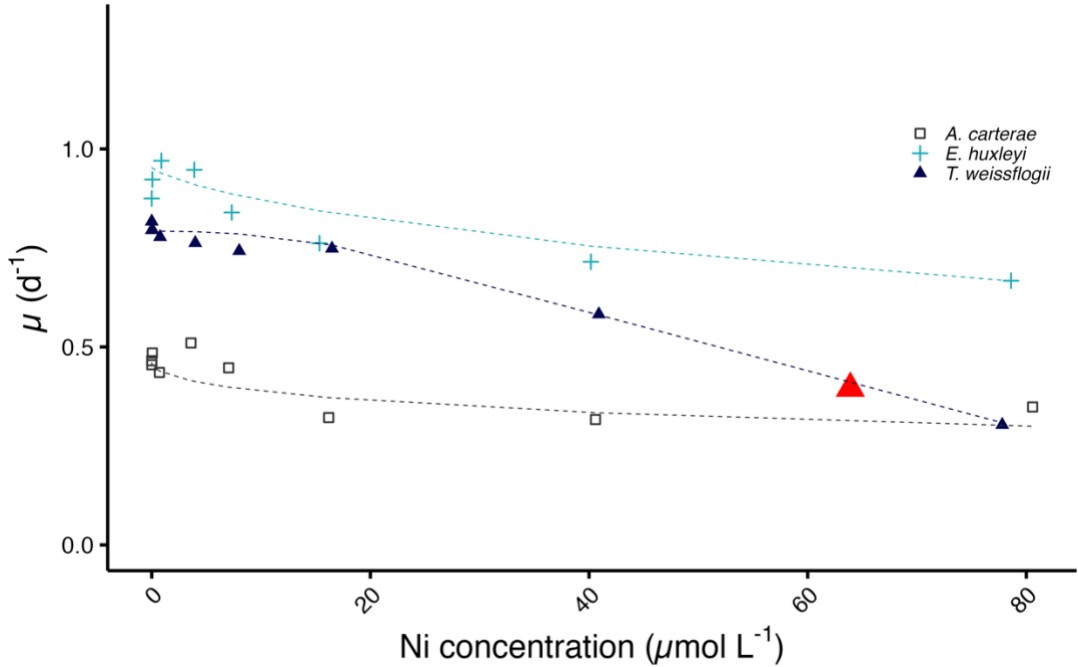


Fig. 5. Predicted growth curves plotted against measured Ni concentrations. The red triangle denotes the IC50 value of *T. weissflogii*. Note that the predicted IC50 values of *A. carterae* and *E. huxleyi* are not shown as they exceed the highest tested Ni concentration.

The diatom *T. weissflogii* has the lowest IC50 value with a concentration of 63.9 µmol L$^{-1}$
while the IC50 values for *A. carterae*, and *E. huxleyi* exceed the highest tested Ni concentration
(Fig. 5).

**4 Discussion**

**4.1 Effects of Ni on marine plankton**

Trace metals are required by phytoplankton for numerous physiological processes and
biochemical reactions; however, it is difficult to disentangle the distinct role of each element.
Ni, for example, is widely recognized to be "bio-required" in several species when urea is
utilized as a nitrogen source (Bartha and Ordal, 1965; Pederson et al., 1986; Price and Morel,
1991). However, to our knowledge, no studies have reported that phytoplankton could benefit
from supplemented Ni when cultivated in nitrate-enriched media while only a few studies have

documented the tolerances of different taxa to progressively increased (possibly toxic) Ni concentrations (Horvatić and Peršić, 2007; DeForest and Schlekat, 2013; Martínez-Ruiz and Martínez-Jerónimo, 2015; Panneerselvam et al., 2018).

Nickel in natural seawater has a concentration lower than 10 nmol $L^{-1}$ (Gerringa et al., 2021; John et al., 2022), and exists mainly in the form of free $Ni^{2+}$ (Donat et al., 1994; Achterberg and Van Den Berg, 1997; Saito et al., 2004). Basic and ultrabasic rocks, which are widely recognized source minerals for OAE, would introduce high amounts of Ni into seawater during mineral dissolution (Renforth, 2019). A wide range of Ni content in olivine (0-0.44 wt%) suggests that the Ni release is source-dependent (Simkin and Smith, 1970). In a previous batch reaction experiment using forsterite olivine sand with 0.26 wt% Ni, an increase of 100 µmol $L^{-1}$ alkalinity was associated with a parallel increase of approximately 3 µmol $L^{-1}$ dissolved Ni during the non-stoichiometric dissolution process (Montserrat et al., 2017). According to these results, the concentration of released Ni could potentially reach the highest concentration tested in this study with a doubling of the current ocean alkalinity level, e.g. at the point source of alkalinity release. In real-world applications, the release of alkaline solutions at discrete locations could potentially lead to "hotspots" of alkalinity and associated increases of Ni in seawater (Bach et al., 2019; Caserini et al., 2021). Alkalinity enhancement modelling studies suggest that the phytoplankton may be impacted by the cumulative effects of alkalinity and released trace metals from recurring local addition (Ilyina et al., 2013, Feng et al., 2017).

In this study, we focused on the impacts of a gradient of Ni concentrations on three key species that belong to different phytoplankton groups. The results showed that while the three tested species were able to survive in all treatments, they displayed adverse responses to high Ni concentrations. The diatom species *T. weissflogii* was the most sensitive species, with an instant reaction to the exposure of Ni and a decrease in cell density when Ni increased to 4.0 µmol $L^{-1}$. At the highest Ni concentration, its growth rate was reduced by 60 %. The dinoflagellate *A. carterae* and coccolithophore *E. huxleyi* are more tolerant to Ni enrichment, with less inhibition in growth rate. The growth rates of *A. carterae* and *E. huxleyi* remained relatively constant beyond a certain threshold and the IC50 values of these two species exceeded the highest tested Ni concentration. The cell densities of *E. huxleyi* were even enhanced when Ni was supplied at low concentrations. Considering the high tolerance *of E. huxleyi* to several other trace metals such as copper and cadmium (Brand et al., 1986), it is not surprising that this species was found to be mostly unaffected by Ni in our study. For example, to counteract high Cu concentrations, *E. huxleyi*, regardless of the needs of the cells, can continuously produce organic Cu-ligand (Echeveste et al., 2018). Another study postulates that *E. huxleyi* survives the Cu stress through

an efficient efflux system by exporting intracellular metals (Walsh and Ahner, 2014). We
speculate that *E. huxleyi* may apply analogous strategies to grow at high nickel concentrations.
Furthermore, Ni was shown to interact with $Ca^{2+}$ and $Mg^{2+}$ transport systems; the uptake of
$Ca^{2+}$ and $Mg^{2+}$ may compete with Ni for the transport pathways and reduce the uptake of Ni in
*E. huxleyi* (Deleebeeck et al., 2009). Interestingly, we observed an enhancement in the cell
densities of *E. huxleyi* at low Ni concentrations. Ni serves as a necessary micronutrient to the
Ni-containing enzyme urease in phytoplankton when the primary nitrogen source is urea (Price
and Morel,1991). However, this does not apply to our study. To the best of our knowledge,
there are no reports indicating the positive effects of nickel as a nutrient when nitrate serves as
the nitrogen source. One possible explanation might be that the introduction of low-dose toxins
prompted an increased rate of cell division, a phenomenon known as hormesis. Studies on
various phytoplankton groups revealed a similar dose-response pattern, where low doses
exhibited beneficial effects and high doses led to toxicity. In these investigations, hormesis was
attributed to low increased levels of Cd (Brand et al., 1986) and Cu (Brand et al., 1986; Pérez
et al., 2006; Yang et al., 2019). This interpretation differs from the notion of metal limitation.
Considering Ni, a slight increase in concentrations positively impacted multiple chlorophyll
fluorescence parameters associated with photosynthesis in terrestrial plants, which was
explained as a hormetic response (Moustakas et al., 2022). Another potential explanation is
that Ni may, to some extent, contribute to the functionality of superoxide dismutase enzymes
which are vital components in an organism's defense against oxidative stress (Sunda 2012).
Either way, this growth alteration should not be dismissed, as it could indirectly impact the
competitive dynamics within ecosystems containing multiple phytoplankton species. Similar
strategies to counteract metal stress were observed in the other species. The dinoflagellate *A.*
*carterae* produces strong ligands to reduce free metal levels (Croot et al., 2000). The growth
rates of *T. weissflogii* were unperturbed about 40 µmol/L and dropped rapidly beyond this
threshold. Production of metal chelators was also reported in diatoms and green algae under
metal stress (Gerringa et al., 1995; Gonzalez-Davila et al., 1995). The release of phytochelatin-
metal complex probably is a detoxification mechanism of the diatom *T. weissflogii* (Lee et al.,
1996). Sequestering metals into a vacuole or storage complexes or binding metals with small
chaperones are also adaptive strategies to buffer the uptake of metals (Blaby-Haas and
Merchant, 2012).
Due to the increasing interest in olivine-based alkalization applications, recent studies
investigated the effects of Ni on marine phytoplankton in the context of OAE. In the study by
Guo et al. (2022), most of the tested phytoplankton species did not exhibit growth inhibition in
response to the tested Ni concentrations, ranging between 0 and 50 µmol L$^{-1}$. The inconsistency
between our results and those of Guo et al. (2022) could be attributed to the different amounts
of bio-available Ni. Guo et al. (2022) utilized a chelator at a high concentration (100 µmol L$^{-1}$
EDTA), while in our experiment 12 µmol L$^{-1}$ EDTA was added. EDTA can chelate free trace
metal ions, forming metal-EDTA complexes. The ligand serves as a buffer by increasing trace
metal availability when trace metal concentrations are low and decreasing trace metal reactivity
at excess levels (Van den Berg and Nimmo, 1987). Several studies documented that
phytoplankton is sensitive to free Ni$^{2+}$ rather than total dissolved Ni (Canterford and Canterford,
1980; Morel et al., 1991; Dupont et al., 2010). Indeed, the lower amount of EDTA employed
in our study led to five orders of magnitude higher concentration of free Ni$^{2+}$ at the target
concentration of 50 µmol L$^{-1}$, compared to that of Guo et al. (2022). We presume that the
variance in the growth inhibition between this study and that of Guo et al. (2022) arises from
the discrepancy of free Ni$^{2+}$ determined by the different amounts of EDTA used in the two
studies. Contrarily, Hutchins et al. (2023) showed that most phytoplankton taxa were
irresponsive to Ni independent of the concentrations of Ni species and EDTA. However, the
study was conducted in a coastal enhanced weathering scenario where the Ni-release process
would be gradual (i.e., years) and the olivine utilized for the experiment contained a low
amount of Ni, measuring 0.13 µmol L$^{-1}$ at the highest concentration. Thus, the synthetic olivine
dissolution yielded lower Ni concentrations, possibly without reaching threshold values of
toxicity compared to those tested in our experiment.
Several studies investigated the impact of Ni at high concentrations on various marine
organisms, albeit outside the specific context of OAE. These studies showed a range of
sensitivities to Ni among different plankton. For example, certain diatom species with low IC50
and copepod species with LC50 (concentration expected to be lethal to 50 % of the tested
organisms) could potentially be vulnerable to the nickel released in the context of OAE. On
the contrary, the lethal concentration of Ni for the dinoflagellate *Prorocentrum donghaiense*
and the diatom *Skeletonema costatum* was found to be 1.7 mmol L$^{-1}$ (Huang et al., 2016), which
is unlikely to be encountered during the process of OAE.
DeForest and Schlekat (2013) suggested a threshold of 20.9 µg L$^{-1}$ Ni (0.35 µmol L$^{-1}$) as the
Predicted No Effect Concentration (PNEC) for chronic Ni toxicity in marine organisms. In a
coastal OAE scenario, with a short water residence time, the low Ni released from alkaline
particles is unlikely to impact the ecosystem due to the slow dissolution rate (Hutchins et al.
2023; Table 2). In the open ocean, olivine must be ground to a very small size (less than 1 µm)
before sinking out of the surface mixed layer (Köhler et al. 2013; Meysman and Montserrat,

2017). Thus, olivine has the potential to release a high quantity of Ni above the IC50 and LC50 values reported in Table 2 for most species. The perturbation could be minimal if the mixing with surrounding waters could rapidly dilute the alkaline solution before impacts in plankton species occur. Therefore, the deployment of alkalinity enhancement in zones with high mixing dynamics could meet the PNEC requirement. Taken together, the introduction of Ni through olivine-based OAE has the potential to shift the taxonomic composition of natural phytoplankton communities. Hence, the observed species-specific sensitivities towards the release of Ni underline that caution is needed in terms of magnitude and temporal mode (e.g., weekly, monthly, seasonal, and annual release) of ocean alkalization to alleviate the cumulative effects of Ni.

Table 2. IC50 and LC50 values ($\mu mol\ L^{-1}$) of different marine organisms. LC50 is the concentration of a material expected to be lethal to 50 % of the tested organisms.

| Taxa | Species | Time (h) | Test water | Ni | IC50 ($\mu mol\ L^{-1}$) | LC50 ($\mu mol\ L^{-1}$) | EDTA ($\mu mol\ L^{-1}$) | Reference |
|---|---|---|---|---|---|---|---|---|
| Diatom | *Odontella mobiliensis* | 96 | Natural | - | 5.28 | | / | (Karthikeyan et al., 2018) |
| Diatom | *Coscinodiscus centralis* | 96 | Natural | - | 10.56 | | / | (Karthikeyan et al., 2018) |
| Diatom | *Skeletonema costatum* | 72 | Natural | - | 154 | | / | (Huang et al., 2016) |
| Diatom | *Phaeodactylum tricornutum* | 72 | Natural | - | 124.04 | | / | (Horvatić and Peršić, 2007) |
| Diatom | *Thalassiosira weissflogii* | 120 | Synthetic | $NiCl_2 \cdot 6H_2O$ | 63.9 | | 12 | this study |
| Dinophyta | *Prorocentrum donghaiense* | 96 | Natural | - | 185 | | / | (Huang et al., 2016) |
| Dinophyta | *Amphidinium carterae* | 120 | Synthetic | $NiCl_2 \cdot 6H_2O$ | >100 | | 12 | this study |
| Cocco-lithophore | *Emiliania huxleyi* | 96 | Synthetic | $NiCl_2 \cdot 6H_2O$ | >100 | | 12 | this study |
| Copepod | *Oithona similis* | 96 | Natural | - | 47.37 | | / | (Karthikeyan et al., 2018) |
| Copepod | *Acartia danae* | 96 | Natural | - | 39.87 | | / | (Karthikeyan et al., 2018) |
| Copepod | *Amphiascus tenuiremis* | 96 | Natural | - | | 11.76 | / | (Hagopian-Schlekat et al., 2001) |
| Copepod | *Tigriopus brevicornis* | 96 | Natural | $NiSO_4 \cdot 6H2O$ | | 3.41 | / | (Barka et al., 2001) |
| Copepod | *Tisbe holothuriae* | 48 | Synthetic | $Ni(CH_3CO_2)_2 \cdot 4H_2O$ | | 44.30 | / | (Verriopoulos and Dimas, 1988) |

**4.2 Implication for the deployment of ocean alkalinity enhancement**

To prevent the potential ecological impacts of Ni in the process of OAE, Ni could be removed during the preparation of alkaline solutions. Numerous techniques that have been employed to remove Ni from wastewater could provide insights into the removal of Ni in olivine (Kadirvelu et al., 2001; Kim et al., 2002; Kalyani et al., 2004; Papadopoulos et al., 2004; Fu et al., 2007; Decostere et al., 2009). For example, chemical precipitation is an effective and the most widely used method in the industry. Though varying success has been achieved, these methods are

associated with high costs, operational drawbacks, and the potential for secondary pollution
(Fu and Wang, 2011). Nowadays Ni is a highly demanded metal resource for battery
manufacture (Turcheniuk et al., 2021). A novel approach has been proposed to recover Ni for
enhancing the supply of critical battery metals (Wang et al., 2018; Wang and Dreisinger, 2023).
This technique could be useful in the context of OAE, contributing to the mitigation of
ecological impacts on the one hand and reducing the costs of OAE application on the other.
For OAE applications, minerals containing less heavy metals, such as quicklime produced from
limestone, could also be considered (Gabe and Rodella, 1999; Šiler et al, 2018). These minerals
could provide a viable option for the required application with less harmful elements
introduced into the ocean (Bach et al., 2019; Caserini et al., 2022). The limestone is abundantly
available and could meet the requirement for large-scale deployment of OAE. In addition, its
economic costs for extraction and transportation are relatively low, and the duration required
for dissolution is shorter compared to olivine (Caserini et al., 2022; Fuhr et al., 2022). However,
it is essential to acknowledge that the calcination of limestone demands a substantial amount
of energy and necessitates proper capture and storage of the released $CO_2$. To comprehensively
assess the applicability and scalability of various material deployments, further investigation
and research are warranted.
**5 Conclusions**
The goal of this study was to examine the response of three phytoplankton species
representative for different taxonomic groups to the exposure of elevated Ni, which may occur
in the process of OAE. The results demonstrated that the tested phytoplankton species exhibited
varying responses to excess Ni. The diatom *T. weissflogii* displayed a high sensitivity to
elevated Ni, evident from its rapid growth inhibition response, high growth inhibition, and low
IC50 value. In contrast, the low growth inhibition and high IC50 values of *A. carterae* and *E.*
*huxleyi* indicate that these two species are more tolerant to excess Ni. The variability in
sensitivity to Ni among different species highlights the importance of avoiding critical toxic
thresholds of Ni concentrations. The recovery of Ni from Ni-rich materials and the usage of
alternative clean minerals would avoid adverse impacts on the phytoplankton community,
enhancing the feasibility and scalability of ocean alkalization. In summary, the varying
responses to Ni among different species make it clear that the impacts of Ni cannot be neglected,
and that caution is needed in setting the threshold for Ni in OAE applications with Ni-rich
materials. Future studies focusing on the taxonomical shift in natural communities and on
incorporation and potential bioaccumulation of Ni in different plankton species are foreseen to
provide a more comprehensive understanding of the potential effects and risks of metal release
associated with OAE.
*Data availability.* The raw data will be made available by the authors, without undue
reservation. The data will be submitted to Pangaea, https://www.pangaea.de/.
*Author contributions.* XX and UR designed the experiment and XX carried them out. XX
conducted statistical analyses and prepared the manuscript with contributions from all authors.
*Competing interests.* The contact author has declared that none of the authors has any
competing interests.

**Acknowledgments**

We gratefully acknowledge the technical support of Mathias Haunost, Tim Steffens in this
study and the invaluable comments and discussions with Fengjie Liu, Jan Taucher and Markus
Schartau on the draft. We also thank Birte Matthiessen and Julia Romberg for kindly providing
the *A. carterae* culture. This study was supported by the German Federal Ministry of Education
and Research (Grant No 03F0895) Project RETAKE, in the framework of the DAM Mission
"Marine carbon sinks in decarbonization pathways (CDRmare). We also acknowledge the
funding from the Carbon to Sea Initiative via the project OCEAN ALK-ALIGN. Xiaoke Xin
is grateful to the China Scholarship Council (CSC) for providing financial support.

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
