# Peer review of "Phytoplankton Response to Increased Nickel in the Context of"

_Biogeosciences, 2023_

## Author Comment (AC1)

We greatly appreciate the valuable comments and critical reading of the manuscript made by the anonymous reviewers, which were useful in improving the scientific quality of the manuscript. Please find below our answers to the Reviewers comments.

RC1: 'Comment on bg-2023-130', Anonymous Referee #1, 19 Sep 2023
Xin et al. measured the cell density of three marine phytoplankton species under different Ni concentrations to assess the influence of Ni on phytoplankton in the context of ocean alkalinity enhancement (OAE). They also measured the total dissolved Ni concentrations and calculated the free Ni concentrations in the growth media to examine the availability or toxicity of Ni. Overall, this study contributes to the current understanding of OAE-related Ni influence. The comments provided below are intended to assist the authors in refining their articles for publication.

**Line 17: Please add a sentence to explain why you chose to study Ni and how it relates to OAE materials.**
Response: Thank you for the above suggestion. We have added information explaining the importance of Ni and its relationship to OAE material. The text has been modified as follows:
*"As one of the most abundant metals in OAE source materials, understanding the impacts of nickel (Ni) on the phytoplankton is critical for the OAE assessment."*

**Line 17: As previously mentioned, Ni can act as a micronutrient at low concentrations. Please consider that dissolved Ni can have both positive and negative effects on phytoplankton growth. Change 'toxicity of nickel' to 'influence/effects of nickel'.**
Response: We rephrased the sentence considering the reviewer's comment.

**Line 18: Change "a range of Ni concentrations" to "9 Ni concentration gradients".**
Response: We changed the text accordingly.

**Line 20 -24: Considering the results revealed some growth enhancement from Ni addition, the fertilization effects of Ni should be mentioned as well. Please write the specific inhibition effects and potential fertilization effects on phytoplankton, such as "XX% of growth rate inhibition" and "IC50 value was XX Ni concentrations".**
**Line 22: The sentence "rapid response to exposure of Ni" is unclear.**
Response: We have broadened the discussion on *E. huxleyi* increase in cell density in chapter 4.1 and have underlined that this growth alteration cannot be dismissed. But in order to keep the flow of the text, we decided not to add this information in the abstract. We have added the detailed concentration of Ni accordingly and modified the text as follows:
*"The impacts of elevated Ni varied among the tested phytoplankton species. The coccolithophore Emiliania huxleyi and the dinoflagellate Amphidinium carterae exhibited a growth rate inhibition of about 30% and 20%, respectively, at the highest Ni concentrations. The half-maximal inhibitory concentration (IC50, at which the growth rate is inhibited by 50%) of both species exceeded the tested range of Ni. This suggests that both species were only mildly affected by the elevated Ni concentrations. In contrast, the diatom Thalassiosira weissflogii displayed a considerably higher sensitivity to Ni, with a 60% growth rate inhibition at the highest Ni concentration and an IC50 value of 63.9 $\mu mol\ L^{-1}$."*

**Line 23: Add "diatom" before "Tholassiosira weissflogii".**
Response: Added to the text.

**Line 58: The word "metal sensitivity" is a wide concept and considering only Ni was investigated in this study, I suggest changing it to "Ni sensitivity".**
 Response: We agree and have changed the text accordingly.

**Line 65: "EDTA", please write out the full name the first time the abbreviation is mentioned.**
 Response: We added the full name of EDTA.

**Table 1. There were standard deviations in Fig.1 so please add the standard deviations in Table 1 for the measured Ni concentrations and potentially free Ni concentrations as well.**
Response: Thank you for the above suggestion. The standard deviations are added in the revised version of this manuscript.

**Line 111-114: Why did the NiCl$_2$ stock solution (line 70) precipitate as nickel carbonate when its maximum solubility is 10.73 mol/L, while the stock solution was only 50 mmol/L? What caused this precipitation of NiCl$_2$?**
Response: This could be attributed to the ability of nickel to form complexes with anionic species rather than chloride, resulting in the formation of compounds (here nickel carbonate) during the invasion of air (Gad, 2023). The solubility of nickel carbonate is 1.58 mmol/L, which is comparatively lower compared to that of NiCl$_2$. This is confirmed by the visual precipitated particles in the stock solution bottle.

*Ref: Gad SC.: Nickel chloride, Reference Module in Biomedical Sciences, Elsevier, https://doi.org/10.1016/B978-0-12-824315-2.00704-1, 2023*

**Fig 2,3,4: It's hard to distinguish the cell density and growth rates in low Ni concentrations (0-5 umol/L). Please consider editing the figure to enhance the differentiation of data points in this low Ni concentration range. It appears that one data point may have been lost or obscured, possibly due to the similarity in results between the control (0 umol/L) and 0.01 umol/L conditions?**
Response: We are aware that some data points are not well-readable due to similar results. All measurements are available in Figures 2, 3 and 4. For Figure 3, due to a technical problem, the measurement of *E. huxleyi* at 0.01 µmol/L was not successful. Thus, the data point was not plotted in the graph. We have specified the missing point in the caption. All the values of the growth rates are now available in Table S1 in the supplementary.

**Discussion: I suggest giving each main discussion paragraph a subtitle. This will help to keep the audience engaged with the clear outlines.**
 Response: Thanks for this comment. We have separated the discussion into two subsections: *"4.1 Effects of Ni on marine plankton; 4.2 Implication for the deployment of ocean alkalinity enhancement."*

**Line 174: Please provide a reference for the sentence "Basic and ultrabasic rocks, which are widely recognized source minerals for OAE, would introduce high amounts of Ni into seawater during mineral dissolution".**
 Response: We added a reference as requested.
*Renforth, P.: The negative emission potential of alkaline materials, Nat. Commun., 10, 1–8, https://doi.org/10.1038/s41467-019-09475-5, 2019.*

**Line 179: (Hartmann et al., 2022) should be (Hartmann et al., 2023).**
 Response: We have corrected it.

**Line 185-209 This paragraph draws comparisons between the responses of three different species based on the results, but it requires some further refinement:**

**Line 193: The influence of Ni at low concentrations (0-5 umol/L) is just as important at high concentrations since the natural Ni concentrations are around 10 nmol/L. It's likely the added Ni from OAE will fall into the range of 0-5 umol/L, and E. huxleyi had enhanced growth rates in this range. Therefore, please discuss why E. huxleyi benefited from supplied Ni at low concentrations.**
  Response: Thank you for the above suggestion. We have added the possible explanation for the enhancement in *E. huxleyi* growth as follows:
*"Considering the high tolerance of E. huxleyi to several other trace metals such as copper and cadmium (Brand et al., 1986), it is not surprising that this species was found to be mostly unaffected by Ni in our study. For example, to counteract high Cu concentrations, E. huxleyi, regardless of the needs of the cells, can continuously produce organic Cu-ligand (Echeveste et al., 2018). Another study postulates that E. huxleyi survives the Cu stress through an efficient efflux system by exporting intracellular metals (Walsh and Ahner, 2014). We speculate that E. huxleyi may apply analogous strategies to grow at high nickel concentrations. Furthermore, Ni was shown to interact with $Ca^{2+}$ and $Mg^{2+}$ transport systems; the uptake of $Ca^{2+}$ and $Mg^{2+}$ may compete with Ni for the transport pathways and reduce the uptake of Ni in E. huxleyi (Deleebeeck et al., 2009). Interestingly, we observed an enhancement in the cell densities of E. huxleyi at low Ni concentrations. Ni serves as a necessary micronutrient to the Ni-containing enzyme urease in phytoplankton when the primary nitrogen source is urea (Price and Morel,1991). However, this does not apply to our study. To the best of our knowledge, there are no clear reports indicating the positive effects of nickel as a nutrient when nitrate serves as the nitrogen source. One possible explanation might be that the introduction of low-dose toxins prompted an increased rate of cell division, a phenomenon known as hormesis. Studies on various phytoplankton groups revealed a similar dose-response pattern, where low doses exhibited beneficial effects and high doses led to toxicity. In these investigations, hormesis was attributed to low increased levels of Cd (Brand et al., 1986) and Cu (Brand et al., 1986; Pérez et al., 2006; Yang et al., 2019). This interpretation differs from the notion of metal limitation. Considering Ni, a slight increase in concentrations positively impacted multiple chlorophyll fluorescence parameters associated with photosynthesis in terrestrial plants, which was explained as a hormetic response (Moustakas et al., 2022). Another potential explanation is that Ni may, to some extent, contribute to the functionality of superoxide dismutase enzymes which are vital components in an organism's defense against oxidative stress (Sunda 2012). Nevertheless, this growth alteration should not be dismissed, as it could indirectly impact the competitive dynamics within ecosystems containing multiple phytoplankton species."*

*Brand, L. E., Sunda, W. G., and Guillard, R. R.: Reduction of marine phytoplankton reproduction rates by copper and cadmium, J. Exp. Mar. Biol. Ecol., 96, 225-250, https://doi.org/10.1016/0022-0981(86)90205-4, 1986.*
*Moustakas, M., Moustaka, J. and Sperdouli, I.: Hormesis in photosystem II: a mechanistic understanding, Curr. Opin. Toxicol., 29, 57-64, https://doi.org/10.1016/j.cotox.2022.02.003, 2022.*

*Price, N. M. and Morel, F. M. M.: Colimitation of phytoplankton growth by nickel and nitrogen, Limnol. Oceanogr., 36, 1071– 1077, https://doi.org/10.4319/lo.1991.36.6.1071, 1991.*

*Pérez, P., Estévez-Blanco, P., Beiras, R. and Fernández, E.: Effect of copper on the photochemical efficiency, growth, and chlorophyll a biomass of natural phytoplankton assemblages, Environ. Toxicol. Chem., 25, 137-143, https://doi.org/10.1897/04-392R1.1, 2006.*

*Sunda, W.G.: Feedback interactions between trace metal nutrients and phytoplankton in the ocean, Front. Microbiol., 3, 204, https://doi.org/10.3389/fmicb.2012.00204, 2012.*

*Yang, T., Chen, Y., Zhou, S. and Li, H.: Impacts of aerosol copper on marine phytoplankton: A review, Atmosphere, 10, 414, https://doi.org/10.3390/atmos10070414, 2019.*

**Line 195-204: If *E. huxleyi* and *A. carterae* produce more organic ligands to decrease the free Ni concentrations, we might observe a decline in free Ni levels at the end of the experiment. Conducting a significance test on the free Ni concentrations among the three species could help determine if the presence of additional ligands reduces Ni toxicity. However, variations in metal quotas among different species introduce additional uncertainty when testing this hypothesis. I recommend that the author revise this discussion.**

Response: The free Ni concentrations were obtained from the Visual MINTEQ 3.1 software calculation and the values are therefore not coming from direct measurements. Since the ligand concentrations are needed as input for the calculations, we cannot provide a significance test for this study to detect the presence of ligands.

**Line 208: The discussion in this section lacks clarity. While the paragraph explains the potential detoxification mechanism in the diatom T. weissflogii, it fails to address why T. weissflogii exhibited higher sensitivity and lower tolerance to high Ni concentrations.**

Response: Thank you for pointing this issue out. Detox mechanisms for *T. weissflogii* are known for this species. However, these studies are based on other trace metals while there isn't any study dedicated to Ni detoxification mechanism. We can therefore only hypothesize that this species applies a similar mechanism for Ni. However, this mechanism does not conflict with *T. weissflogii* sensitivity to Ni, which resulted in being higher compared to the other two tested species. We rephrased the text to clarify. "*The growth rate of T. weissflogii was unperturbed until about 40 µmol/L and dropped rapidly beyond this threshold.*"

**Line 212: The cited reference Guo et al., (2022) used different Ni concentrations (0-50 umol/L) and gradients, so please write the specific Ni concentrations compared here to avoid potentially misleading information. Change "in response to high Ni concentrations" to specific Ni concentrations**.

Response: We added the specific Ni concentrations: "*In the study by Guo et al. (2022), most of the tested phytoplankton species did not exhibit growth inhibition in response to the tested Ni concentrations, ranging between 0 and 50 µmol L$^{-1}$.*"

**Line 220: "…in our study led to orders of magnitude…" how many orders of magnitude?**

Response: The number of orders is added. "*Indeed, the lower amount of EDTA employed in our study led to five orders of magnitude higher concentration of free Ni$^{2+}$ at the target concentration of 50 µmol L$^{-1}$ compared to that of Guo et al. (2022).*"

**Line 227: "… contained a low amount of Ni…" what's the concentration of Ni used in Hutchins et al. (2023)?**

Response: The specific concentration is added. *"However, the study was conducted in a coastal enhanced weathering scenario where the Ni-release process would be gradual (i.e., years) and the olivine utilized for the experiment contained a low amount of Ni, measuring 0.13 µmol $L^{-1}$ at the highest concentration."*

**Line 231: "These studies showed a range of sensitivities to Ni among different groups." Please change the word "groups" to a more specific description, like "plankton" etc.**

Response: Thank you for the suggestion. The word has been changed to plankton.

**Line 231: "LC 50" Please explain or at least write out the full name when the abbreviation first appears.**

Response: We rephrased the sentence and added an explanation for LC50 as follow: *"For example, certain diatom species with low IC50 and copepod species with LC50 (concentration expected to be lethal to 50 % of the tested organisms), could potentially be vulnerable to the nickel released in the context of OAE (see Table 2)."*

**Line 234: I suggest using the IC50 instead of the "lethal concentration" if you have the information because in this study IC50 was calculated and discussed.**

Response: IC50 is not an appropriate indicator for larger organisms with low growth rates. We have rephrased the sentence to remove ambiguity as a former reply.

**Line 235: Move "(Huang et al., 2016)" after the "1.7 mmol/L".**

*Response:* We have moved the reference accordingly.

**Line 229-239: Considering the information presented in Table 2 and the potential impact of nickel addition mentioned in line 176, it is advisable for the authors to delve deeper into the discussion of a nickel threshold in the context of OAE projects.**

Response: We took into account the reviewer's comment. The discussion has been expanded accordingly.

*"DeForest and Schlekat (2013) suggested a threshold of 20.9 µg $L^{-1}$ Ni (0.35 µmol $L^{-1}$) as the Predicted No Effect Concentration (PNEC) for chronic Ni toxicity in marine organisms. In a coastal OAE scenario, with a short water residence time, the low Ni released from alkaline particles is unlikely to impact the ecosystem due to the slow dissolution rate (Hutchins et al. 2023; Table 2). In the open ocean, olivine must be ground to a very small size (less than 1 µm) before sinking out of the surface mixed layer (Köhler et al. 2013; Meysman and Montserrat, 2017). Thus, olivine has the potential to release a high quantity of Ni above the IC50 and LC50 values reported in Table 2 for most species. The perturbation could be minimal if the mixing with surrounding waters could rapidly dilute the alkaline solution before impacts in plankton species occur. Therefore, the deployment of alkalinity enhancement in zones with high mixing dynamics could meet the PNEC requirement. Taken together, the introduction of Ni through olivine-based OAE has the potential to shift the taxonomic composition of natural phytoplankton communities. Hence, the observed species-specific sensitivities towards the release of Ni underline that caution is needed in terms of magnitude and temporal mode (e.g., weekly, monthly, seasonal, and annual release) of ocean alkalization to alleviate the cumulative effects of Ni."*

*DeForest, D. K. and Schlekat, C. E.: Species sensitivity distribution evaluation for chronic nickel toxicity to marine organisms, IEAM, 9, 580-589, https://doi.org/10.1002/ieam.1419, 2013.*

*Köhler, P., Abrams, J.F., Völker, C., Hauck, J., and Wolf-Gladrow, D.A.: Geoengineering impact of open ocean dissolution of olivine on atmospheric $CO_2$, surface ocean pH and marine biology, Environ. Res. Lett., 8, 014009, https://doi.org/10.1088/1748-9326/8/1/014009, 2013.*

*Meysman, F.J. and Montserrat, F.: Negative $CO_2$ emissions via enhanced silicate weathering in coastal environment, Biol. Lett., 13, 20160905, https://doi.org/10.1098/rsbl.2016.0905, 2017*

**Line244: Please add a reference for the sentence "Nowadays Ni is a highly demanded metal resource for battery manufacture".**

Response: Thank you for the above suggestion. We added the following reference:
*Turcheniuk, K., Bondarev, D., Amatucci, G.G. and Yushin, G.: Battery materials for low-cost electric transportation, Mater. Today, 42, 57-72, https://doi.org/10.1016/j.mattod.2020.09.027, 2021.*

**Line 253: Limestone is not a metal-free resource mineral (Šiler, 2018). In fact, nearly all minerals used in OAE contain metals, including elements like Ca and Mg. Please change it into a more accurate description.**

Response: We have rephrased the sentence following the reviewer's suggestion as: *"For OAE applications, minerals containing less heavy metals, such as quicklime produced from limestone, could also be considered (Gabe and Rodella, 1999; Šiler et al, 2018)."*

*Gabe, U. and Rodella, A. A.: Trace elements in Brazilian agricultural limestones and mineral fertilizers, Commun. Soil. Sci. Plant. Anal., 30, 605–620,* https://doi.org/10.1080/00103629909370231, *1999.*

*Šiler, P., Kolářová, I., Bednárek, J., Janča, M., Musil, P. and Opravil, T.: The possibilities of analysis of limestone chemical composition, IOP Conf. Ser.: Mater. Sci. Eng., 379, 012033,* https://doi.org/10.1088/1757-899X/379/1/012033, *2018.*

**Line 253-257: The use of limestone differs significantly from that of olivine. Limestone cannot be employed directly as an OAE material, making the example provided somewhat misleading. A revision of this discussion is necessary to provide clarity and context for the subsequent discussion on energy consumption.**

Response: Thank you for the above suggestion. We have changed limestone to quicklime so that the application of olivine and quicklime are comparable.

**Please review and ensure the accuracy of the reference format, including the inclusion of all necessary information for each citation. Specifically, please make sure to use subscript "2" for "$CO_2$," superscript "2+" for "$Mg^{2+}$," avoid capitalizing journal titles, provide DOI numbers, and use standard abbreviations for journal names.**

Response: Thank you for the above suggestion. We have revised the reference format and ensured the format meets the requirements of Biogeosciences.

References:

---

## Author Comment (AC2)

**RC2**: , Anonymous Referee #2, 04 Oct 2023 reply

The manuscript by Xin et al. presents results on the response of important phytoplankton to increased Ni concentrations in the context of olivine-based ocean alkalinity enhancement. This study is novel, timely, and the methods, analyses, and interpretation seem to be scientifically sound. I recommend publication, but kindly ask the authors to consider the following questions and comments when revising their manuscript.

**Line 39: "…co-benefit of mitigating ocean acidification…" – state that this would especially occur at OAE deployment sites, prior to seawater equilibration with $CO_2$.**

 *Response:* Thank you for the above suggestion. We agree that OAE has the co-benefit of mitigating OA at the deployment sites. However, we respectfully disagree that this happens prior to seawater equilibration with $CO_2$ as the pH is still a bit higher than the pre-OAE application levels after equilibration. We modified the text accordingly. *"OAE, in addition to enhancing the buffering capacity of seawater, has the co-benefit of mitigating ocean acidification at the deployment sites (Köhler et al., 2010)."*

**Lines 41-42: Consider adding a brief discussion regarding the merits of using olivine over other proposed OAE minerals here.**

*Response:* We revised the text*: "Among the most recognized alkaline minerals, olivine rocks have gained considerable attention due to their relative fast weathering rate, wide availability, and low cost (Schuiling and Krijgsman, 2006; Hartmann et al., 2013)."*

**Lines 55-56: Why did the authors select these three organisms for their study? Are they meant to be functional group representatives? If so, a brief note regarding the importance of the functional groups they represent would be useful.**

Response: We provided a brief introduction to explain their importance as follows: "*In this study, we examined the impacts of Ni on three representative marine phytoplankton species: the diatom Thalassiosira weissflogii, the dinoflagellate Amphidinium carterae, and the coccolithophore Emiliania huxleyi. These species were selected as they represent the three dominant functional groups of phytoplankton."*

**Lines 70-80: This section would benefit from the inclusion of additional information. What were the experimental timeframes and why were they selected? Although timeframes are shown in Figs 2c, 3c, and 4c, discussing them here would be helpful.  The experimental timeframe for the *A. carterae* experiment appears to be two days longer than that of the *E. huxleyi* and *T. weissflogii* experiments – what was the reason for this? For how many generations was each species grown? The replication strategy is also not clear to me. Were samples collected from triplicate independent replicates at each sampling point? Finally, I would suggest editing this section's heading – the heading implies that methods used for determining species responses will be presented, but it seems to focus more on the experimental design; section 2.3 seems to focus more on methods used to determine species response.**

Response: Thank you for the above suggestion. We rephrased the text accordingly. The changes are reported in the following lines:

*Line 83: We have changed the heading of 2.2 to "Experimental setup" to remove the overlap with the heading of 2.3.*

*Lines 85: Experiments were performed in triplicate independent 75 mL falcon flasks for each of the Ni concentrations.*

*Line 87: Samples (1 mL) of each flask were collected into sterile 2 mL microtubes.*

*Lines 89–95: To minimize the impact of changes in the carbonate chemistry of the medium induced by cellular metabolism, the phytoplankton biomass at the harvest time should*

*consume less than 5% of the total dissolved inorganic carbon (Zondervan et al., 2002). The maximum density was determined based on the cell carbon quota of the tested species. Accordingly, in this study, the maximum cell densities of A. carterae, T. weissflogii, and E. huxleyi never exceeded 12000 cells mL$^{-1}$, 15000 cells mL$^{-1}$, and 130000 cells mL$^{-1}$, respectively (Zondervan et al., 2002; Olenina et al., 2006).*

*Olenina, I., Hajdu, S., Edler, L., Andersson, A., Wasmund, N., Busch, S., Göbel, J., Gromisz, S., Huseby, S., Huttunen, M., Jaanus, A., Kokkonen, P., Ledaine, I., and Niemkiewicz, E.: Biovolumes and size-classes of phytoplankton in the Baltic Sea HELCOM Balt, Sea Environ. Proc. No. 106, 144 pp., 2006.*

*Zondervan I., Rost B. & Riebesell U.: Effect of CO$_2$ concentration on the PIC/POC ratio in the coccolithophore Emiliania huxleyi grown under light-limiting conditions and different daylengths, J. Exp. Mar. Biol. Ecol., 272, 55-70,* https://doi.org/10.1016/S0022-0981(02)00037-0, *2002.*

**Lines 75-76: "…between 9:00 a.m. and 10:00 a.m.…" – to avoid the effect of the photocycle…**

Response: We have rephrased the sentence based on the suggestion as follows*: "Samples were always collected at the same time of day between 9:00 a.m. and 10:00 a.m. to avoid an effect of the photocycle."*

**Lines 94-95: Consider rewording this sentence for clarity and to define IC50 (e.g. – "The inhibitory concentration [IC50], at which growth is inhibited by 50%, was…").**

Response: We have rephrased the sentence as follows*: "The half maximal inhibitory concentration (IC50), at which the growth rate is inhibited by 50%, was determined from the dose-response curve."*

**Table 1: As the measured and free Ni$^{2+}$ concentrations presented in Table 1 are from triplicate measurements, standard deviations be included here**.

Response: The standard deviations are now provided in Table 1 in the revised version of the manuscript.

**Figs. 2a-b, 3a-b, and 4a-b: The data points are a bit hard to read, especially at lower Ni concentrations.**

Response: We are sorry for this inconvenience. We added a table (Table S1) with all growth rate values in the supplementary.

**Fig. 2c, 3c, 4c: Data points are missing error bars. Additionally, units should be provided for log-transformed cell densities**.

Response: The error bars are always available. If not visible, error bars are smaller than symbols. We specified this in the caption of the graphs.

The units have been added as cell density (log10 cells mL$^{-1}$).

**Lines 139-140: "A similar trend…" – this sentence does not flow well.**

Response: We have rephrased the sentence as follows: *"The cell densities and growth rates of E. huxleyi increased with the addition of Ni up to 3.89 μmol L$^{-1}$. At 0.9 μmol L$^{-1}$ Ni, the maximum cell density with a 63 % increase was observed (p < 0.01). The growth rate increased by about 11 %, but this value is not statistically significant (p = 0.07)."*

Line **143-144: I find this interpretation a bit confusing: the authors note that *E. huxleyi* growth was inhibited after day 3 upon being exposed to Ni, pointing to Fig. 3c, but – in**

**some treatments – log cell densities appear to be higher than the control after day 3. Clarifying which treatment level is being referred to would reduce ambiguity.**

Response: Thank you for the above suggestion. We have rephrased this paragraph as follows: *"With the increase in Ni concentration after 15.3 μmol L$^{-1}$ Ni, the density started to decrease and the growth rate was inhibited. From 15.3 to 78.6 μmol L$^{-1}$ Ni, the cell densities decreased significantly between 57-72 % compared to the control (p < 0.05; Fig. 3a). The decrease in growth rate reached up to 24 % at the highest Ni concentration compared to the control (Fig. 3b). The growth variance of E. huxleyi started on day 3 after being exposed to Ni (Fig. 3c)."*

**Line 153-154: Here, too, clarification about which treatment level is being referred to would be helpful.**

Response: We have rephrased the paragraph as follows: *"The cell densities of T. weissflogii remained relatively stable until 8.0 μmol L$^{-1}$ Ni, after which the densities started to decrease significantly (p < 0.05; Fig. 4a). Similarly, the growth rates of T. weissflogii remained relatively stable until 40.9 μmol L$^{-1}$ Ni, above which the growth rates started to decrease significantly (p < 0.05; Fig. 4b). After being exposed to Ni, T. weissflogii reacted immediately from day 2 onwards (Fig. 4c)."*

**Line 164: Please consider removing "Up to today,". Consider the following as a potential alternative: "Trace metals are required by phytoplankton for numerous physiological processes and biochemical reactions; however, it is difficult to disentangle the distinct role of each element."**

Response: We have rephrased the sentence following the reviewer's suggestion.

**Line 168: Wouldn't "nutrient-enriched media" be a better choice here since f/2 media was used?**

Response: We agree that f/2 media could be considered as nutrient-enriched media and changed the "nitrate-enriched" to "nutrient-enriched".

**Lines 193-200: The authors should also discuss the observed increase in *E. huxleyi* specific growth rates at low Ni$^{2+}$ concentrations. What are the proposed hypotheses that explain this increase?**

Response: Thank you for the above suggestion. We have added the possible explanation for the enhancement in *E. huxleyi* growth also based on another reviewer's comment as follows: *"Interestingly, we observed an enhancement in the cell densities of E. huxleyi at low Ni concentrations. Ni serves as a necessary micronutrient to the Ni-containing enzyme urease in phytoplankton when the primary nitrogen source is urea (Price and Morel,1991). However, this does not apply to our study. To the best of our knowledge, there are no clear reports indicating the positive effects of nickel as a nutrient when nitrate serves as the nitrogen source. One possible explanation might be that the introduction of low-dose toxins prompted an increased rate of cell division, a phenomenon known as hormesis. Studies on various phytoplankton groups revealed a similar dose-response pattern, where low doses exhibited beneficial effects and high doses led to toxicity. In these investigations, hormesis was attributed to low increased levels of Cd (Brand et al., 1986) and Cu (Brand et al., 1986; Pérez et al., 2006; Yang et al., 2019). This interpretation differs from the notion of metal limitation. Considering Ni, a slight increase in concentrations positively impacted multiple chlorophyll fluorescence parameters associated with photosynthesis in terrestrial plants, which was explained as a hormetic response (Moustakas et al., 2022). Another potential explanation is that Ni may, to some extent, contribute to the functionality of superoxide dismutase enzymes which are vital components in an organism's defense against oxidative*

*stress (Sunda 2012). Nevertheless, this growth alteration should not be dismissed, as it could indirectly impact the competitive dynamics within ecosystems containing multiple phytoplankton species."*

*Brand, L. E., Sunda, W. G., and Guillard, R. R.: Reduction of marine phytoplankton reproduction rates by copper and cadmium, J. Exp. Mar. Biol. Ecol., 96, 225-250, https://doi.org/10.1016/0022-0981(86)90205-4, 1986.*
*Moustakas, M., Moustaka, J. and Sperdouli, I.: Hormesis in photosystem II: a mechanistic understanding, Curr. Opin. Toxicol., 29, 57-64, https://doi.org/10.1016/j.cotox.2022.02.003, 2022.*
*Price, N. M. and Morel, F. M. M.: Colimitation of phytoplankton growth by nickel and nitrogen, Limnol. Oceanogr., 36, 1071– 1077, https://doi.org/10.4319/lo.1991.36.6.1071, 1991.*
*Pérez, P., Estévez-Blanco, P., Beiras, R. and Fernández, E.: Effect of copper on the photochemical efficiency, growth, and chlorophyll a biomass of natural phytoplankton assemblages, Environ. Toxicol. Chem., 25, 137-143, https://doi.org/10.1897/04-392R1.1, 2006.*
*Sunda, W.G.: Feedback interactions between trace metal nutrients and phytoplankton in the ocean, Front. Microbiol., 3, 204, https://doi.org/10.3389/fmicb.2012.00204, 2012.*

*Yang, T., Chen, Y., Zhou, S. and Li, H.: Impacts of aerosol copper on marine phytoplankton: A review, Atmosphere, 10, 414, https://doi.org/10.3390/atmos10070414, 2019.*

**Line 226-227: If space permits, it would be useful to discuss the range of Ni concentrations found in olivine, especially within the context of the Ni concentrations selected for this study (Simkin and Smith [1970] could be useful for this).**
Response: Thank you for the above suggestion. We have included the discussion regarding the range of Ni concentrations. We added the discussion in lines 204-212 for a better continuity as follows: *"Basic and ultrabasic rocks, which are widely recognized source minerals for OAE, would introduce high amounts of Ni into seawater during mineral dissolution (Renforth, 2019). A wide range of Ni content in olivine (0-0.44 wt%) suggests that the Ni release is source-dependent (Simkin and Smith, 1970). In a previous batch reaction experiment using forsterite olivine sand with 0.26 wt% Ni, an increase of 100 µmol L$^{-1}$ alkalinity was associated with a parallel increase of approximately 3 µmol L$^{-1}$ dissolved Ni during the non-stoichiometric dissolution process (Montserrat et al., 2017). According to these results, the concentration of released Ni could potentially reach the highest concentration tested in this study with a doubling of the current ocean alkalinity level, e.g. at the point source of alkalinity release."*

*Simkin, T. and Smith, J. V.: Minor-element distribution in olivine, J. Geol., 78, 304–325, https://doi.org/10.1086/627519, 1970.*

**Line 231: LC50 should be defined here.**
Response: We rephrased the sentence and added an explanation for LC50 to remove the ambiguity.
*"For example, certain diatom species with low IC50 and copepod species with LC50 (concentration expected to be lethal to 50 % of the tested organisms), might be susceptible to the released nickel in the context of OAE (see Table 2)."*

**Lines 252-253: Limestone and its derivatives are unlikely to be metal-free (e.g., Gabe and Rodella, 1999).**
Response: We have rephrased the sentence following the reviewer's suggestion: *"For OAE applications, minerals containing less heavy metals, such as quicklime produced from limestone, could also be considered (Gabe and Rodella, 1999; Šiler et al, 2018)."*

**Line 256: Consider including a note about the slow dissolution times observed for olivine as well (e.g., Fuhr et al., 2022).**
Response: We added the discussion regarding the slow dissolution time of olivine as follows: *"In addition, its economic costs for extraction and transportation are relatively low, and the duration required for dissolution is shorter compared to olivine (Caserini et al., 2022; Fuhr et al., 2022)."*
*Fuhr, M., Geilert, S., Schmidt, M., Liebetrau, V., Vogt, C., Ledwig, B., and Wallmann, K.: Kinetics of olivine weathering in seawater: an experimental study, Front. Clim., 4, 39, https://doi.org/10.3389/fclim.2022.831587, 2022.*
Lines 262-273: Based on their results, what do the authors suggest for future experiments?
Response: We added the discussion based on the results as follows: *"Future studies focusing on the taxonomical shift in natural communities and on incorporation and potential bioaccumulation of Ni in different species under cumulative Ni are foreseen to provide a more comprehensive understanding of the potential effects and risks of metal release associated with OAE."*

---

## Author Response (AR2)

Dear editor,

We would like to thank you for your suggestion. We deleted the repeating point that Ni serving as a micronutrient when the primary nitrogen source is urea. We moved the text regarding hormesis downwards as a separate paragraph for the readability of the text. We also deleted the point that the cell densities of *E. huxleyi* were enhanced in line 229 in the manuscript containing tracked changes. This appeared to be a repeating point of hormesis hypothesis. From line 193–281, there are 4 paragraphs in total in the manuscript containing tracked changes.

With regards,

On behalf of the co-authors, Xiaoke Xin